# The Automated Test of Embodied Cognition: Concept, Development, and Preliminary Findings

**DOI:** 10.3390/brainsci13060856

**Published:** 2023-05-25

**Authors:** Morris David Bell, Alexander J. Hauser, Andrea J. Weinstein

**Affiliations:** 1Department of Psychiatry, Yale University School of Medicine, New Haven, CT 06510, USA; alexander.hauser@yale.edu (A.J.H.);; 2VA Connecticut Healthcare System, 950 Campbell Avenue, West Haven, CT 06516-2770, USA

**Keywords:** embodied cognition, neuropsychology, substance use disorder, alcohol use disorder, aging, cognitive assessment

## Abstract

(1) Background: The Automated Test of Embodied Cognition (ATEC) uses video administration of cognitively demanding physical tasks and motion capture technology to assess cognition in action. Embodied cognition is a radical departure from conventional approaches to cognitive assessment and is in keeping with contemporary neuroscience. (2) Methods: ATEC was administered to a convenience sample of 20 patients with substance use disorder and 25 age-matched community controls. Patients were administered concurrent cognitive assessments. (3) Results: Psychometric analysis revealed excellent internal consistency, test–retest reliability and small practice effects. Groups were significantly different on ATEC scores and ATEC scores significantly related to concurrent measures of cognition. (4) Conclusions: The preliminary results support the reliability and validity of ATEC for older adults.

## 1. Introduction

The purpose of this article is to present a new neurocognitive assessment system based on the construct of embodied cognition called The Automated Test of Embodied Cognition (ATEC). We describe its theoretical background, its development as a digital assessment system, and preliminary findings regarding its reliability and validity.

### 1.1. Overview of ATEC

The Automated Test of Embodied Cognition (ATEC) measures cognitive functioning based on cognitively demanding physical tasks assessed using an automated video administration with motion capture technology. Embodied cognition (EC) is a broad framework within cognitive science that emphasizes the importance of somatic sensorimotor experiences with our social and physical environment in developing and shaping our higher cognitive processes [1]. It argues that biological aspects of bodily life are necessary for cognition, affecting our perceptions and our interpretations of experience. Developmental psychology recognizes the key role that sensorimotor information plays in cognitive development and conceptual processes [2], and there is ample evidence that cognitive decline is associated with reduced physical activity and a lack of sensorimotor stimulation [3]. ATEC addresses the need for an assessment instrument that measures the integration of sensorimotor and neurocognitive functioning as a measure of higher cognitive processes in action. 

ATEC was the result of a collaboration between computer scientists and psychiatrists, neuropsychologists, neurologists and physical therapists to develop an objective measurement system of embodied cognition for children with a range of neurodevelopmental stages and disorders, funded by a large grant from NSF to Morris Bell, Ph.D, as Co-PI, with a computer science team headed by Fillia Makedon, Ph.D, of the University of Texas, Arlington (National Science Foundation Project #1565310, “Cyber Human Systems Large Collaborative Research: Computational Science for Improving Assessment of Executive Function in Children”). The collaboration produced an automated assessment system with motion capture technology [4] that has been found to be reliable and valid for the assessment of higher cognitive functioning in children. It is a better predictor of day-to-day functioning in children than conventional neuropsychological measures, and differentiates normal children from children at risk of neurodevelopmental disorders [5]. An adult version, with similar assessment tasks and technology but with adult demonstrators and more adult language, has been developed using many tasks from the child version, thus providing a system which can assess people throughout life, from 5 years to 90 years of age. 

### 1.2. Embodied Cognition (EC): A Novel Construct to Study Sensorimotor/Cognitive Impairment

Sensorimotor and neurocognitive compromise are among the earliest indicators of onset of illness in neurodevelopmental disorders and neurocognitive decline, and they are predictors of functional outcomes [6,7]. Neuropsychological assessments have long been the usual procedures for identifying these impairments, but these tests come from early 20th century understandings of the mind as being separate from the body, and contemporary cognitive testing methods, like the first IQ test (Stanford Binet), remain seated tasks with very little demand for body movement. ATEC measures EC, a cognitive neuroscience framework that addresses critical relationships between cognition and body movement in response to structured activities, in ways that may elude conventional sensorimotor or neurocognitive assessments.

In brief, EC proposes that cognition is shaped by the entire body system. Neuroscience reveals that cognition develops both along with and by way of physical movement [2]. The brain is topographically organized, with multiple parallel basal ganglia–cortical circuits (i.e., motor, limbic, cognitive circuits), so that even higher cortical functions such as working memory and self-regulation (prefrontal cortex) are actually part of complex distributed systems that include cerebellar and subcortical regions previously associated only with movement and balance. 

While neuropsychological assessments are anchored in a strong foundation of cognitive science, their procedures are based on disembodied and localizationist–connectionist approaches. Clinical tests rely on classic models of perception–cognition–action that ignore the important sensorimotor system and emotions, while sensorimotor tasks (such as neurological subtle signs (NSS)) do not make higher cognitive demands. Measuring cognition in action is a paradigm shift that is in keeping with contemporary neuroscience. Higher level neurocognitive assessments that require movement are needed, because EC tasks are truer to real-life functions (i.e., they have ecological validity) and capture cognition in action. EC tasks can provide more sensitive assessments because of the greater neurocognitive challenge posed when the whole body is engaged; during EC tasks, higher cognitive functioning is presented in the context of sensorimotor processing, as is the case with dual attention tasks used in neurological assessments [8]. 

### 1.3. Utilizing Novel Digital Approaches to Detect Sensorimotor/Cognitive Impairment

Measuring cognition in action is only now practical because of advances in motion capture technology that make automated administration and scoring possible. The field of digital health technology, which uses a variety of platforms, is growing at a rapid pace and emerging in studies of various neurological conditions [9,10]. Digital technology has become a notable research priority, as evidenced by the 2018 National Academies of Sciences, Engineering, and Medicine’s Forum on Neuroscience and Nervous System Disorders, which addressed the field’s current states and challenges, and the translation of digital technology into meaningful health and societal contributions [10]. With ATEC, we have created an automated video administration and a comprehensive scoring system with high fidelity, usability, and efficiency. The motion capture setup uses inexpensive, commercially available equipment, is portable and is user friendly. ATEC incorporates an engaging platform, in which the participant interacts with a recorded “host” who is aided by demonstrators representing gender, age, and racial diversity. Tasks are administered on screen (similarly to an exercise video) ensuring high fidelity and reliability of administration, while allowing users the flexibility to personalize the assessment by selecting tasks appropriate to the reason for referral.

ATEC is a new tool with the potential for detecting sensorimotor/cognitive change during early stages of illness or over the course of recovery by offering novel metrics rather than improvements to older tests [11]. This technology promises to provide sensitive, objective, multidimensional and ecologically valid measurements that reflect subtle changes in behavior or function; thus, it has the potential to produce clinically meaningful outcome measures in clinical practice and research trials. We also have plans for developing a telehealth version for home-based monitoring.

This article focuses on the reliability and validity of ATEC measurement using expert scoring. The specific aims were to establish the test–retest reliability, discriminate validity, and concurrent validity using a subsample who had undergone neurocognitive assessments. These are preliminary data and will need replication with larger samples of adults.

## 2. Material and Methods

### 2.1. Participants

Participants were from a convenience sample of participants enrolled in substance abuse clinical research (n = 20; 70% alcohol use disorder, 25% polysubstance use disorder and 5% opioid use disorder) and from age-matched community controls (n = 25) for a combined sample of N = 45 [12]. Retesting of ATEC was performed on 23 participants (10 community controls and 13 patients) who were available and agreed to return for a second assessment. The total sample mean age was 53.6 years (range 21 to 89), and groups did not differ significantly by age (t = 1.22, df = 43, *p* = 0.230). The total sample was 73.9% male and 60.0% White, 24.4% African American, 8.9% Asian, and 8.9% other. 

### 2.2. ATEC

The adult version uses many of the same original executive functioning tasks as the child version (marching, red light/green light/yellow light, and cross your body, as described below), but contains the tasks from Section 3 of the Movement Disorder Society-Unified Parkinson’s Disease Rating Scale (MDS-UPDRS), as well as a dual attention task and an embodied learning and delayed memory recall task, called map sense, which was added because of the importance of delayed recall in assessing mild cognitive impairment. 

The MDS-UPDRS movement tasks include timed-up and go (stand up from a sitting position and walk ten feet, turn around and walk back), tandem walk (walk eight steps heel to toe), the Romberg task (stand with eyes closed and arms outstretched for 10 s), balance on one foot (for ten seconds), foot tap, foot stomp, fist open and close, hand pronation–supination, and finger tap (all tasks repeated for left and right for 10 s). The dual attention task combines timed-up and go with counting backwards from 100 by 7′s. This shows the difference in walking speed with and without cognitive load. 

There are four novel tasks that were developed specifically for ATEC. One of these is marching in place at a slow and fast tempo as a measure of rhythm and coordination without complex cognitive load. To test higher cognitive processes related to executive functions (EF), two novel physical tasks were added to the standard neurological tasks. As previously described in our article on the child version of ATEC [5], one is an attention and response inhibition task called “red light/yellow light/green light”. It is similar to computerized continuous performance tests (CPT; e.g., [13]) that assess sustained attention and response inhibition, but is more complex and requires rhythmic upper body movement in response to commands. The participant is asked to pass a juggling ball from one hand to the other in time with the words “green light”, to move the ball up and down in time with the words “yellow light”, and to not pass the ball when the participant hears “red light”. The task is subsequently repeated at a faster pace. The participant is then presented with the same task, but using a sequence of pictures of green, red, and yellow traffic lights as visual cues, rather than the spoken cues, thus allowing for comparison between sensory modalities. 

The second EF task is the “cross your body” task. In beat to a tune, participants are instructed to touch their ears alternately with the opposite hand (left hand to right ear; right hand to left ear) three times, and then the knees three times (lyrics: “cross your body, touch your ears, ears, ears; cross your body, touch your knees, knees, knees”). Then, participants are instructed to touch their knees when the word ears is heard, and touch their ears when the word knees is heard. The same procedure then replaces ears and knees with hips and shoulders. In the final round, all four commands are given, and the person must remember to touch their knees when ears is heard, their ears when knees is heard, their hips when shoulders is heard, and their shoulders when hips is heard. This task requires sustained attention, working memory, response inhibition, cognitive flexibility, and self-regulation. Crossing the midline increases sensitivity for detection of subtle brain compromise and increases cognitive load.” 

The fourth novel task is an embodied learning and memory task called “map sense”. It requires the participant to remember three, four, and then five step movements across a 3 × 3 grid. The steps are sequentially displayed on a map on screen, and then the participant must remember the sequence and move across the grid on the floor in rhythm to a simple tune. There are three trials for each map. The participant is tested again 20 min later without the maps being shown to measure delayed embodied memory recall. A list of the ATEC tasks is shown in Table 1 along with the cognitive demands they represent. Expert scoring creates raw scores at the item level (e.g., number of finger taps, accurate ball passes), and these raw scores are then categorized so that each component score of each task (e.g., accuracy, rhythm) has equal value. These are added together into domain scores and converted again into a 5-point scale. The domain scores are then added together to give the ATEC total score. Expert scoring was evaluated for interrater agreement when developing the child version; it exceeded ICC < 0.96 for the raw scores, and there was near perfect agreement for converted scores. Scoring for the adult version was carried out using the consensus of at least two raters. The scoring template for expert scoring is provided in Appendix A, which also shows the order of task administration. 

### 2.3. Neurocognitive Assessments

The Montreal Cognitive Assessment (MoCA [14] and the MoCA Memory Index Score (MIS) provide an assessment of global cognitive functioning and verbal delayed recall. Scores were adjusted using published age and education norms [15]. The Neuropsychological Assessment Battery (NAB) Mazes [16] test was used to assess EF with no working memory. Mazes was the neuropsychology test selected by NIMH as the best single measure of EF for their MATRICS Consensus Cognitive Battery (MCCB), developed for clinical trials. 

### 2.4. Procedures

Participants provided written informed consent according to procedures approved by the VA Connecticut Healthcare System Institutional Review Board. They completed neurocognitive assessments and self-report questionnaires according to manual instructions and were then administered the ATEC. ATEC set-up involves a conventional computer with its monitor and a large screen monitor with a conventional webcam placed on top. The ATEC facilitator uses the computer monitor to push a button that starts the instructions for each task that appears on the large monitor for the participant. Once the participant understands the instructions, the facilitator pushes a second button that initiates the task administration on the large screen, and simultaneously starts the webcam, which automatically ends at the conclusion of the task. The recording is stored on the hard drive and the facilitator then moves to the next task. The user-friendly interface allows the facilitator flexibility if a shorter examination is desired (for example, only EF tasks), but for this study, all tasks were included. For balance tasks, the facilitator stands nearby the participant for safety. ATEC is as safe as any standard clinical neurological examination, and no adverse events of any kind have occurred with any participant. The ATEC’s administration time is approximately 45 min. 

After the ATEC, some participants (n = 16) who enrolled in a study involving cognitive assessments as well as ATEC were given a questionnaire about their experience of ATEC. The ATEC Participant Experience Questionnaire has ten questions to be answered on a Likert scale of 1 to 5, from strongly disagree to strongly agree (see Appendix A). The maximum score is 50 points. Several questions have reverse scoring to balance any response bias. They were also administered the MoCA and NAB Mazes. 

### 2.5. Scoring

ATEC scoring was performed by experts who previously achieved excellent inter-rater agreement (intra-class r = 0.96). They followed ATEC scoring guidelines (Appendix A). Scoring was carried out using the consensus of at least two raters. These scores are used as the “ground truth” for the development of scoring algorithms (which is ongoing) [17,18,19]. The raw scores were categorized, and the categorical scores were converted on a 5-point scale into “converted scores”, so that each task has equal weight for determining the domain scores. The ATEC total score is the sum of the eight domain scores.

### 2.6. Analysis

Cronbach’s alpha was used to determine the internal consistency of the eight ATEC domain scores. Internal consistency was not analyzed within domains, although we anticipate doing so when we have a larger data set. Intra-class correlation was used to determine test–retest reliability; an ANCOVA, independent sample *t*-tests and binary logistic regression were used for comparing community controls and patients. A receiver operator characteristic (ROC) was used to determine the classification sensitivity of the ATEC total score to group membership. Pearson correlations were used for associations between relevant ATEC domains and other cognitive testing. All tests were two-tailed, and the alpha was set to 0.05.

## 3. Results

### 3.1. ATEC Internal Consistency

The ATEC internal consistency for the eight domain scores was excellent, with a Cronbach’s alpha of 0.869. Delayed embodied memory recall was the only domain identified that would very slightly increase Cronbach’s alpha if removed (0.879). This indicates that this domain may be capturing something less related to the other domains than they are to each other.

### 3.2. ATEC Test–Retest Reliability and Practice Effects

Test–retest reliability (n = 23) at approximately 2 weeks (median = 14 days) was excellent, with an intraclass r of 0.936, *p* < 0.001. There was only a small practice effect. The mean ATEC total score at visit one was 32.26 (4.97), and two weeks later was 33.57 (5.85), showing a difference of 1.31 and a small Cohen’s d’ = 0.24. When community controls were removed, the patients (n = 13) showed almost no improvement. The mean difference between visit 1 (mean = 30.08 (4.31)) and visit 2 (30.77 (5.31)) was 0.69, representing a very small Cohen’s d’ = 0.14. 

### 3.3. ATEC Discriminant Validity

ATEC total score was the dependent variable with age as a covariate in an ANCOVA comparing the patient sample (n = 20) and age-matched community controls (n = 25). Since age was found to be related to a decline in the ATEC total score (r = −0.59, *p* < 0.001), as would be expected if the ATEC is sensitive to age-related cognitive decline, it was included in the model; however, age was not significantly different between groups (community controls mean = 57.20 (16.36), patient mean = 50.76 (18.63), t =1.22, df = 43, *p* = 0.23). The community control ATEC total score mean of 35.08 (4.25) was significantly higher than the patient sample mean = 30.60 (4.62); F (2,1) = 19.24, *p* < 0.001, partial eta squared = 0.476, adjusted R-squared = 0.453. Age was a significant covariate (F (2,1) = 21.57, *p* < 0.001). ROC analysis using ATEC total score produced an area under the curve = 0.79, and excellent sensitivity was achieved, with a cut-off score of 30.9. 

Since ATEC total score showed a robust between groups effect, we attempted to explore which domains might be contributing most to this result. First, we performed independent *t*-tests to identify likely candidates and found that with Bonferroni correction for eight comparisons (*p* = 0.05/8 = 0.00625), two domains had significant effects (response inhibition: t (43) = 3.38, *p* = 0.002; balance: t (43) = 3.30, *p* = 0.002). These were entered into a binary logistic regression, with age in the first block, and response inhibition and balance in the second. This produced a significant model (chi-sq = 11.53, df = 2, *p* < 0.003) and had a classification accuracy of 80%. These results are similar to what was found for ATEC total score, suggesting that these two domains may be most responsible for the group difference found for ATEC total score.

### 3.4. ATEC Concurrent Validity (Relationship to Neurocognitive Testing)

For a subgroup of patients who received cognitive assessments (n = 16), significant associations were found between ATEC total score and MoCA total (r = 0.627, *p* = 0.012). A non-significant association was found between MoCA MIS and ATEC total score (r = 0.447, *p* = 0.095). However, when MIS was compared to the two ATEC memory domains, working memory (r = 0.671, *p* = 0.006, Bonferroni corrected for two comparisons (*p* = 0.05/2 = 0.025) showed a significant relationship, although delayed memory was not related (r = 0.16, *p* = ns). Significant associations were also found between ATEC total score and NAB Mazes, a measure of EF that does not involve working memory (total r = 0.564, *p* = 0.023). Correlations between NAB Mazes and the domains related to executive function (self-regulation, response inhibition and attention) reveal significant relationships for all three domains (self-regulation (r = 0.651, *p* = 0.006), attention (r = 0.636, *p* = 0.008), and response inhibition (r = 0.631, *p* = 0.009), with Bonferroni correction for three comparisons (*p* = 0.05/3 = 0.0167)). 

### 3.5. ATEC Participant Experience

Participants (n = 16) reported very favorable experiences with ATEC (an overall mean score of 46.4/50). Most reported that they found the tasks challenging (13 agree or strongly agree), safe (16 agree or strongly agree), relevant to their medical and physical health (15 agree or strongly agree), and comfortable (16 agree or strongly agree), and would not hesitate to do the assessment again (15 agree or strongly agree). 

## 4. Discussion

ATEC presents a novel approach to cognitive assessment that embraces contemporary neuroscience understandings of the relationship between cognition and physical movement. It is composed of cognitively demanding physical tasks representing essential neurocognitive domains. In a previously published paper, the child version of ATEC was found to be psychometrically sound and to have concurrent, discriminant and predictive validity, and was more sensitive to parents’ reports of their child’s functioning than standard neurocognitive assessments. For the adult version of ATEC, the balance and motor speed tasks are adopted from the MDS-UPDRS, while the executive function-related tasks are original. Many of these tasks (marching, red light/green light, cross your body) are the same as the child version, allowing for the potential of repeated assessments throughout life. Embodied learning and memory recall were added specifically for assessing memory loss in early dementia. This is the first published report on the internal consistency, test–retest reliability, practice effects, discriminant validity, concurrent validity, and patient acceptability of the adult ATEC. 

Results show that ATEC has strong internal consistency and excellent test–retest reliability at two weeks. Importantly, only a small practice effect was found for the entire sample, and almost no practice effect was found for the patient sample. This finding means that when assessing the effects of interventions, even small improvements in performance may be attributable to the intervention rather than to practice effects.

As expected, age was found to be inversely related to performance for the entire sample, which is the opposite of what we found in the child version, wherein scores increased with age, as would be expected with normal neurodevelopment. With adequate samples, we will be able offer age-adjusted norms across the lifespan, but until then, we wanted to use age as a covariate in comparing our community control and patient group, even though the groups were not significantly different in age. We found strong support for discriminant validity with the community control sample having an ATEC total mean score significantly higher than that of the patient group. This is especially interesting because all patients were community-dwelling, functional adults who did not report any comorbid neurological problems. They were in outpatient treatment for alcohol or substance us disorders and were not particularly aware of cognitive difficulties. Yet, ATEC easily differentiated the two groups with fairly high sensitivity, according to the ROC analysis. An exploratory analysis of domain scores found that response inhibition and balance were the two domains that most significantly differed between the two groups, and with similar classification accuracy, suggesting that these two domains were primarily responsible for the ATEC total score finding. Response inhibition is conceptually related to addictive disorders, but balance was unexpected. It may be that this represents early cerebellar signs of the effects of many years of severe alcohol use. Motor speed, however, did not differ between groups, and may be less affected by substance use disorders than other cognitive features.

Concurrent validity with a small patient subsample showed remarkably robust relationships of ATEC total score with MoCA and with Mazes. ATEC working memory was robustly associated with MoCA’s memory index score, and ATEC’s EF domains of self-regulation, response inhibition, and attention were strongly related to Mazes, which measures executive function. This analysis was seriously underpowered, but these findings provide tentative support for the validity of these domains. 

Participants rated the ATEC experience as safe, challenging, and relevant to their medical and physical health. Most found it enjoyable and would not hesitate to do it again. The ATEC takes approximately 45 minutes to complete, which is much shorter than a standard neuropsychological assessment of the same cognitive domains. The ATEC was tolerated well by all participants. 

There are a number of limitations to the present study. Most notable is the sample size, which meant that only very large effects reached significance. The fact that discriminant and concurrent analyses reached statistical significance is an indication of the robustness of the findings. However, this initial study used only two standard neurocognitive assessments, and there were no self-reported measures of functional impairment or cognitive complaints. Future studies will include a broader range of assessments that will help us understand relationships between ATEC domain scores and their conceptually related standard cognitive tests.

The patient group was a convenience sample participating in substance abuse research and was not seeking treatment for cognitive complaints. The advantage of this sample is that their impairments were subtle, yet the ATEC clearly differentiated them from age-matched community controls. The ATEC is intended to be used whenever neurocognitive assessments are needed in clinical practice, but we expect it to be especially useful in detecting the early stages of neurodegenerative disorders (e.g., multiple sclerosis, Parkinson’s disease, all cause dementia), and for tracking recovery from stroke, TBI, and psychiatric disorders including schizophrenia, mood disorders, substance abuse and PTSD. Its low practice effects and high test–retest reliability make it especially suitable for tracking the longitudinal course of disease, rehabilitation, and recovery, and as a measurement instrument for clinical trials.

In recent years, there has been a growing body of research regarding the relevance of EC to treatment mechanisms and outcomes in psychotherapy [20,21]. The ATEC offers psychotherapy researchers and practitioners a highly accessible tool for assessing psychotherapeutic outcomes related to EC, circumventing the prohibitive cost and time associated with conventional neurocognitive assessment.

ATEC was developed to be an automated assessment system that includes video administration and will soon have automated scoring. Expert scoring, the “ground truth” for algorithm development, was used for analysis in this article. Automated scoring was developed using motion capture data and deep machine learning algorithms to generate scores that matched the expert scoring. Algorithms achieved 99% agreement with expert scores for cross your body and red light/yellow light/green light, the two most essential tasks for EF [17,18,19]. Algorithm development for other tasks is ongoing. Since ATEC uses conventional computer equipment including a typical webcam, it is potentially a highly transferrable technology which can be easily installed and used in clinical settings. A prototype version is in development and will be available soon. We also plan to develop a version of ATEC adapted to telehealth, so that ATEC can be used in remote settings and potentially for home monitoring. The user-friendly interface makes it possible to select only those tasks needed to respond to a given clinical concern, so that shorter and more focused ATEC assessments are possible. The preliminary findings presented in this article support the continued development of ATEC. If these findings hold up with larger and more diverse samples, ATEC may become an important addition to clinical evaluations and research by providing an assessment that reflects contemporary understandings of the relationship between body and mind.

## Figures and Tables

**Table 1 brainsci-13-00856-t001:** ATEC cognitive demands, tasks and scoring domains (* MDS-UPDRS; ** higher cognitive function).

Type	Tasks	Domains
* Gross motor and gait, proprioception, vestibular	Timed up and go, dual attention, tandem gait, Romberg, standing on one foot (L, R)	Balance
** Rhythmic movement	Marching slow and fast	Rhythm/Coordination
** Embodied learning and memory	3, 4, 5 step maps	Working memory, Rhythm/Coordination
** Embodied memory and delayed recall	20 min delayed recall	Delayed recall
* Bilateral coordination	Ball pass to the beat slow and fast	Rhythm/Coordination
** Response inhibition	Red light/yellow light/green light slow and fast/auditory and visual	Response inhibition, Rhythm/Coordination, Attention
** Bilateral coordination and self-regulation	Cross your body	Self-regulation, Rhythm/Coordination, Working memory
* Rapid sequential movements	Foot tap, foot stomp, fist open and close, hand pronate/supinate, finger tap (all L, R)	Motor speed

## Data Availability

Deidentified data are available from the author upon approval from the VA Privacy and Information Security offices.

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
