# Peer review of "The Automated Test of Embodied Cognition: Concept, Development, and Preliminary Findings"

_brainsci, 2023, doi:10.3390/brainsci13060856_

Round 1

Reviewer 1 Report

GENERAL COMMENTS:

Line 330: …deep learning algorithms to generate scores – Does this mean that there is no standardized way of scoring? Does the algorithm always regenerate scores in a different manner? Scoring should be explained in more detail for all tasks, although the authors mention it, it is not sufficient (the reader should know more about it in order to understand the findings). Although a scoring guide is added as a supplement I believe that explaining it in 2-3 sentences in the text would improve the paper's clarity.

Timed Up and Go, Dual Attention, Tandem Gait, Rom-berg, Standing on One Foot, Foot Tap, Foot Stomp, Fist Open and Close, Hand Pronate/Supinate, Finger Tap – These tasks should be explained at least shortly in the text.

Why did authors use substance abuse groups? Any references which would support such a decision and argumentation in line with that?

Are groups matched by gender too, or some other important characteristics? If not, authors can use measures that they already have (gender and cognitive measures) to show additional group equality or to argue for possible cognitive deficits (cognitive measures).

I would suggest analyzing the data by hierarchical binary logistic regression, with the group as a dependent, age and cognitive tests as predictors in the first step (model), and ATEC scores in the second step. This way authors can show incremental validity of ATEC scores and avoid multiple testing (Type I error inflation).

SPECIFIC COMMENTS

Line 63: EC is the theory – Hypothesis, the theory is a broader concept.

Line 72: perception-cognition-action ignore the important sensory-motor system and emotions while sensorimotor tasks – Such tasks exist (for example Object Motion Tracking paradigm) but they are mainly used in research (perception) and not as standardized tests in practice.

Line 99: technology can provide sensitive, objective, multidimensional and ecologically valid measurements that reflect subtle – References are needed for these claims since this is the paper that aims to investigate ATEC characteristics. I think it is too early to state strong statements like these in the introduction, maybe to say that so far, on children it did show good characteristics…

Line 54: The fourth novel task is a embodied memory task, called “Map Sense – this is not mentioned previously when authors listed all tasks, so please just check if task names are matched during the listing and later while describing.

Line 196: The ATEC Participant Experience Questionnaire – Can you give an item example here?

Line 209: ATEC Internal Consistency for the 8 Domain scores was excellent – This is the reliability for the total score, not for all domains? Are there alpha values for each domain also (if each domain constitutes out of several scores)?

Line 214: Test-Retest reliability (n=23) - In the procedure part no retest was mentioned and it should be, how were the participants chosen, was the task order randomized or not…

Line 233: independent t tests were performed for each domain – Since 8 t-tests were performed, at least Bonferroni correction should be applied, so the significance level should be 0.05/8=0.00625, and accordingly only the first two domains would be significant. A comment is missing for these results.

Line 238: ATEC Concurrent Validity – since 18 correlations were tested here (8 domains + total with two cognitive measures), according to Bonferroni correction, a significance level should be 0.05/18=0.002778, according to which none of the correlations is significant.

Line 250: Participants (n =16) reported very favorable experiences – Why did not all participants fill in this questionnaire?

Line 320: clinical staff yet were clearly revealed with ATEC – We cannot be sure that ATEC detected cognitive impairments here, it might be that it detected something else that it also captures. So a milder conclusion should be made here. 

Author Response

Point by point reply to Reviewer 1 comments

Thank you for these helpful suggestions; below please find point by point responses.

GENERAL COMMENTS:

  1. Line 330: …deep learning algorithms to generate scores – Does this mean that there is no standardized way of scoring? Does the algorithm always regenerate scores in a different manner? Scoring should be explained in more detail for all tasks, although the authors mention it, it is not sufficient (the reader should know more about it in order to understand the findings). Although a scoring guide is added as a supplement I believe that explaining it in 2-3sentences in the text would improve the paper's clarity.

Reply: A section has been added to procedures to describe the scoring process.  The scores reported in this article are not algorithm produced scores.  They are the expert scores.  They are the “ground truth” for algorithms which are under development.

  1. Timed Up and Go, Dual Attention, Tandem Gait, Romberg, Standing on One Foot, Foot Tap, Foot Stomp, Fist Open and Close, Hand Pronate/Supinate, Finger Tap – These tasks should be explained at least shortly in the text.

Reply: I have added more detail about each task.

  1. Why did authors use substance abuse groups? Any references which would support such a decision and argumentation in line with that? Are groups matched by gender too, or some other important characteristics? If not, authors can use measures that they alreadyhave (gender and cognitive measures) to show additional group equality or to argue for possible cognitive deficits (cognitive measures).

Reply: As I stated in the text, the substance abuse group was a convenience sample. We have a line of research in our lab related to cognitive impairment as a barrier to recovery from substance use disorders. I argue that this sample poses a special challenge for detecting group differences because these patients are not seeking assessment or treatment for cognitive impairment and are not particularly aware of their cognitive problems.  Therefore, ATEC findings of discriminant validity with this sample suggest that more obviously impaired samples (e.g., Parkinson’s Disease, MCI, TBI) would show even bigger effects. I have added a reference related to our research on cognitive impairment in substance abuse.

  1. I would suggest analyzing the data by hierarchical binary logistic regression, with the group as a dependent, age and cognitive tests as predictors in the first step (model), and ATEC scores in the second step. This way authors can show incremental validity of ATEC scores and avoid multiple testing (Type I error inflation).

Reply: I would love to have done that analysis.  Unfortunately, we don’t have cognitive assessment data on the control sample. What I could do and did was to take the suggested approach in regard to looking at domain scores by entering Age in Block 1 and Domain Scores for those domains that showed t-test differences between groups (Bonferroni corrected).  This does not answer the question of incremental validity, but it does deal with the problem of Type 1 error inflation.

SPECIFIC COMMENTS

  1. Line 63: EC is the theory – Hypothesis, the theory is a broader concept.

Reply: I have changed the sentence to say EC proposes.

  1. Line 72: perception-cognition-action ignore the important sensory-motor system and emotions while sensorimotor tasks – Such tasks exist (for example Object Motion Tracking paradigm) but they are mainly used in research (perception) and not as standardized tests in practice.

Reply: I have changed the sentence to more specifically state: Clinical tests rely on classic models of perception-cognition-action that ignore the important sensory-motor system and emotions while sensorimotor tasks (like Neurological Subtle Signs (NSS)) do not make higher cognitive demands.

  1. Line 99: technology can provide sensitive, objective, multidimensional and ecologically valid measurements that reflect subtle – References are needed for these claims since this is the paper that aims to investigate ATEC characteristics. I think it is too early to state strong statements like these in the introduction, maybe to say that so far, on children it did show good characteristics…

Reply: I agree that it is too early to make strong claims. I have changed the language to: “This technology promises to provide sensitive, objective, multidimensional and ecologically valid measurements…”

  1. Line 54: The fourth novel task is a embodied memory task, called “Map Sense – this is not mentioned previously when authors listed all tasks, so please just check if task names are matched during the listing and later while describing.

Reply: Thank you for point this out.  When we introduce the new tasks, I have now added:  “and an embodied learning and delayed memory recall task, called Map Sense, which was added because of the importance of delayed recall in assessing mild cognitive impairment.

  1. Line 196: The ATEC Participant Experience Questionnaire – Can you give an item example here?

Reply: I have added the questionnaire to the Supplementary materials (Appendix B).

  1. Line 209: ATEC Internal Consistency for the 8 Domain scores was excellent – This is the reliability for the total score, not for all domains? Are there alpha values for each domain also (if each domain constitutes out of several scores)?

Reply: We have not yet looked at each Domain’s internal consistency, so I do not have that analysis for this paper. I have added this statement: “Internal consistency was not analyzed within Domains, although we anticipate doing so when we have a larger data set.” 

  1. Line 214: Test-Retest reliability (n=23) - In the procedure part no retest was mentioned, and it should be, how were the participants chosen, was the task order randomized or not.

Reply: We have now added the retest to the procedure section, and in the Participant section we specify the number of community controls and patients in the retest subsample. Task order is not randomized.  The order is fixed.  We have no reason to believe that the order of the testing is a factor.  Participants can take breaks between tasks if there is evidence of fatigue, but so far we have not encountered any problem with patients tolerating the demands of the tasks.

  1. Line 233: independent t tests were performed for each domain –Since 8 t-tests were performed, at least Bonferroni correction should be applied, so the significance level should be0.05/8=0.00625, and accordingly only the first two domains would be significant. A comment is missing for these results.

Reply: We have added Bonferroni Corrections to these tests as suggested.

  1. Line 238: ATEC Concurrent Validity – since 18 correlations were tested here (8 domains + total with two cognitive measures) according to Bonferroni correction, a significance level should be0.05/18=0.002778, according to which none of the correlations is significant.

Reply: We understand the problem with multiple comparisons. So given this concern, we used a hypothesis driven approach. After testing for ATEC Total Score with MoCA, MIS and NAB Mazes, we selected only those Domains that logically should be associated with MoCA MIS (the two memory Domains) and NAB Mazes (the three EF Domains). We used Bonferroni correction separately because there were separate hypotheses. The consequence of this hypothesis driven approach was that we deleted the association we had found between NAB Mazes and Rhythm.  Therefore, that finding and its discussion no longer appear in this report.

  1. Line 250: Participants (n =16) reported very favorable experiences– Why did not all participants fill in this questionnaire?

Reply: We have explained this in the Participant section. This was a procedure added with one of the substance abuse protocols.  We are continuing to collect data, of course, but think that in this preliminary report, we should provide our findings to date.

  1. Line 320: clinical staff yet were clearly revealed with ATEC – We cannot be sure that ATEC detected cognitive impairments here, it might be that it detected something else that it also captures. So a milder conclusion should be made here.

Reply: We have deleted and rewritten the statement to be milder.

Reviewer 2 Report

The paper describes a novel test based on EC theory to measure cognition. The paper is well-written and introduces an important topic. Several changes should be considered before publications. As the title includes the specification preliminary the methodology is appropriate for preliminary findings. 

First minor corrections such in abstract such as 1) (2)

Insert abbreviation for EC earlier in the paper

First the paper is presented as a report. Either do the changes within the article or suggest in the title that is a report e.g. The purpose of this report …please replace with the purpose of this article….

…..the construct of embodied cognition. Please respect the consistency of the term…and use instead the theory/framework of embodied cognition. Also instead of the model use the term theory/framework. In the case of presenting the model of EC applied to summarize EC for formulation of the test, use model but it needs to be described. If not, use consistently the term theory or better framework of embodied cognition (since are many theories under EC umbrella concept).

e.g., line 59 ATEC measures Embodied Cognition (EC) - a cognitive neuroscience 59 model that ad

Line 28 add body since the main focus of both EC and the paper is on body…. It emphasizes the importance of the body, sensorimotor experiences with  our social and physical environment in developing and shaping our higher cognitive…

Please insert citation in the beginning  for EC Varela, F. J., Thompson, E., & Rosch, E. (1991). The embodied mind: Cognitive science and human experience. The MIT Press.

Line 74 please remove underlining ….Measuring cognition in action is a paradigm shift, in keeping with contemporary neuroscience.

Use for p three digits p = .23).

Remove dots due to self correction… line 174 age and education [14].

Line 214 insert spaces (n=23)

Line 296 replace our…….. our executive function measure. …which measures executive function.

Line 332 please rephrase … We expect that a prototype will soon be available from the author

Line 325 after PTSD extend the applications to psychotherapy

As EC theory has been included to guide psychotherapy interventions (Tiba & Manea, 2018), including the battery into psychotherapy assessment has also the potential to enrich the conceptualisation, provide a guidance for EC based psychotherapy interventions, and to measure the rehabilitation effect of psychotherapy on EC outcomes  (Nyman-Salonen, Kykyri, & Penttonen, 2022)

Nyman-Salonen P, Kykyri VL, Penttonen M. Challenges and added value of measuring embodied variables in psychotherapy. Front Psychiatry. 2022 Dec 16;13:1058507. doi: 10.3389/fpsyt.2022.1058507. PMID: 36590641; PMCID: PMC9800897.

Tiba, A. & Manea, L. (2018). An Embodied Simulation Account of Cognition in Rational Emotive Behavior Therapy. New Ideas in Psychology 48 C pp. 12-20 doi: 10.1016/j.newideapsych.2017.08.003

Author Response

Point by point reply to Reviewer 2 comments­­

Thank you for your favorable review and very helpful notes. Please find point-by-point responses to your comments below.

  1. First minor corrections such in abstract such as 1) (2)

Reply: In line 6 I changed 1) to (1) for consistency.

  1. Insert abbreviation for EC earlier in the paper

Reply: In line 27 I introduced the abbreviation EC at first mention of Embodied Cognition. In line 65 I replaced “embodied cognition” with just the abbreviation EC.

  1. First the paper is presented as a report. Either do the changes within the article or suggest in the title that is a report e.g. The purpose of this report …please replace with the purpose of this article….

Reply: In line 20, the word report is replaced with article. In line 116, report is replaced with article.

  1. …the construct of embodied cognition. Please respect the consistency of the term…and use instead the theory/framework of embodied cognition. Also instead of the model use the term theory/framework. In the case of presenting the model of EC applied to summarize EC for formulation of the test, use model but it needs to be described. If not, use consistently the term theory or better framework of embodied cognition (since are many theories under EC umbrella concept).

Reply: In line 27, concept is changed to framework. In line 65, model is changed to framework.

  1. Line 28 add body since the main focus of both EC and the paper is on body ….

Reply: “…the importance of sensorimotor experiences…” is changed to “…the importance of somatic sensorimotor experiences…”

  1. Please insert citation in the beginning for EC Varela, F.J., Thompson, E., & Rosch, E. (1991). The embodied mind: Cognitive science and human experience. The MIT Press.

Reply: Citation added in line 30 after first use of the term embodied cognition.

  1. Line 74 please remove underlining …Measuring cognition in action is a paradigm shift, in keeping with contemporary neuroscience.

Reply: Underlining removed.

  1. Use for p three digits p = .23).

Reply: Third digit added to reflect p = .230.

  1. Remove dots due to self correction… line 174 age and education.

Reply: This was corrected to “…age and education norms.”

  1. Line 214 insert spaces (n=23)

Reply: Everywhere that the formatting (n=x) appeared without spaces, spaces were inserted.

  1. Line 296 replace our ……. our executive function measure. … which measures executive function.

Reply: I changed “our executive function measure” to “which measures executive function”.

  1. Line 332 please rephrase … We expect that a prototype will soon be available from the author.”

Reply: I changed this sentence to “A prototype version is in development and will be available soon.”

  1. Line 325 after PTSD extend the applications to psychotherapy …..

Reply: A paragraph was added to the discussion section addressing potential applications of ATEC in psychotherapy.

Reviewer 3 Report

Review of The Automated Test of Embodied Cognition - concept, development, and preliminary findings

The paper describes a novel means to assess cognition in action, The Automated Test of Embodied Cognition (ATEC), through video administration and analysis of motion capture data, extending earlier promising findings from children to older adults. Psychometrics show good internal consistency, test-retest reliability and minimal practice effects, revealing significantly differences between cognitively impaired adults and an age-matched control group from the community.

As a learning scientist who conducts research and engages in classroom based intervention designs to understand the benefits of embodied learning, I found this research to be highly informative for advancing both theory and practice. I do have some methodological and theoretical concerns with this manuscript that I hope will help to advance the scientific footing of this work and its reliable application in clinical and educational settings.

GENERAL COMMENTS

My theoretical concern is to ask whether the particular forms of integration of sensorimotor and neurocognitive functioning are the same as what learning scientists generally regard as "embodied cognition" and “grounded and embodied learning?” I would like to see the authors address the nature of this relationship more explicitly.

One methodological concern I have concerns the balance tasks of the ATEC.  It is understandable that the facilitator stands nearby for safety during the balance tasks. However, this kind of social interaction may have implications for people's cognitive and motoric behavior, such as the social facilitation- inhibition effect (Zajonc, 1965, 1968). How often and in what ways was support provided? Did this show influences on participant performance? I would also like to know if there are considerations to revise these tasks to provide supports without a person, such as having participants position other forms of balance support with inanimate objects such as a wall or railing that are not subject to social interactions that might influence behavior.

My other methodological concern is about the data analysis. Performing multiple t-tests between identical groups as was done for each domain has been shown to produce high false positive rates. This is because the chances of making a type I error for multiple comparisons is greater than the error rate for a single comparison. Best practices dictate one should use ANOVAs and follow-up methods such as Tukey’s method or the Bonferroni’s correction for multiple comparisons. I would be far more comfortable with the stated results if these proper statistical practices were employed, since I feel the current analyses do call the current reported results into question.

My final point is directed at the Discussion. It is promising that this assessment detected cognitive impairments that apparently went unnoticed by clinical staff. It raises a question whether clinical staff could be trained to notice these cognitive impairments better by attending to certain types of motor behaviors outside of a formal assessment. Do the authors think these are trainable to human observation?

SPECIFIC COMMENTS

Note references 1 & 7 appear to be identical (Borghi & Cimatti 2010)

Please review the text on pp. 3-4 since it is confusing which novel task is intended to be first, second, third, fourth. For example, the fourth task is named immediately after the second task.

Author Response

Reply to Reviewer 3

We thank this reviewer for his favorable review and his comments to improve the manuscript.

GENERAL COMMENTS

  1. My theoretical concern is to ask whether the particular forms of integration of sensorimotor and neurocognitive functioning are the same as what learning scientists generally regard as "embodied cognition" and “grounded and embodied learning?” I would like to see the authors address the nature of this relationship more explicitly.

Reply: The reviewer has made an excellent suggestion, pointing out that there are several silos of investigation into this broad topic.  My background and the focus of this paper is embedded in the domain of medical science. I would love to read a review article that tries to parse the differences proposed by this general comment, but I do not feel equipped to write such an article.  The focus of this paper is on the development of a medical device in keeping with the acute need to improve neurocognitive assessments for people at-risk or who have neurocognitive compromise.  I hope the reviewer can understand that this paper’s narrow focus on that objective precludes the theoretical exploration that the reviewer proposes. 

  1. One methodological concern I have concerns the balance tasks of the ATEC. It is understandable that the facilitator stands nearby for safety during the balance tasks. However, this kind of social interaction may have implications for people's cognitive and motoric behavior, such as the social facilitation- inhibition effect (Zajonc, 1965, 1968). How often and in what ways was support provided? Did this show influences on participant performance? I would also like to know if there are considerations to revise these tasks to provide supports without a person, such as having participants position other forms of balance support with inanimate objects such as a wall or railing that are not subject to social interactions that might influence behavior.

Reply: Support is only provided for the balance tasks which come directly from the Movement Disorder Society-Unified Parkinson’s Disease Rating Scale (MDS-UPDRS).  This is the standard for neurological exams and requires that the safety of the patient be ensured by having someone available to catch the person if they might stumble or fall.  To do anything else other than to have a person right there would be a potential violation of safety protocol and not allowed in medical settings. It is highly unlikely that this safety measure influences the task performance which is about balance.  Higher cognitive tasks in the study do not have someone standing close by so there should be no such influence.  It is of course possible that a grumpy administrator or an especially enthusiastic or empathic administrator may have some small effect, but this would be true in any neuropsychological testing session. We would like to point out that since the administration of the instructions and tasks are standardized because they are presented as a video, the administrator effect is likely reduced as a source of variance.

  1. My other methodological concern is about the data analysis. Performing multiple t-tests between identical groups as was done for each domain has been shown to produce high false positive rates. This is because the chances of making a type I error for multiple comparisons is greater than the error rate for a single comparison. Best practices dictate one should use ANOVAs and follow-up methods such as Tukey’s method or the Bonferroni’s correction for multiple comparisons. I would be far more comfortable with the stated results if these proper statistical practices were employed, since I feel the current analyses do call the current reported results into question.

Reply: We agree with this recommendation and have provided Bonferroni corrections.  Reviewer 1 had similar concerns and we have addressed these more fully in our reply to Reviewer 1.

  1. My final point is directed at the Discussion. It is promising that this assessment detected cognitive impairments that apparently went unnoticed by clinical staff. It raises a question whether clinical staff could be trained to notice these cognitive impairments better by attending to certain types of motor behaviors outside of a formal assessment. Do the authors think these are trainable to human observation?

Reply: First, I have removed the reference to clinical staff observations in the discussion at the suggestion of reviewer 1.

Secondly, Clinical observation is extremely important and is a central feature of medical education. In fact, most neurological exams amount to the examiner observing the patient for a few minutes doing various tasks and asking them to count backwards from 100 by 7’s or to remember 3 words.  The problem of course is that there is low fidelity to these methods and no formal scoring. MDS-UPDRS was an example of trying to raise the practice to a higher level of science by having a systematic scoring system. ATEC takes this approach much further by providing a high-fidelity administration and a more refined scoring system. ATEC then goes on to measure executive functions.

Still, I am aware that ATEC is not measuring everything that a clinician might note and regard as important. For example, we are not measuring tremor or sway.  There are other sophisticated systems for these measurements and medical examiners immediately recognize these signs of illness. We hope someday to use our motion capture data to add measurements of these physical signs, but mostly we have chosen in this first iteration of ATEC to focus on higher cognitive functions which are not so observable. It is unlikely that a clinician could make valid conclusions from observations related to higher executive functioning except in the most general terms.

SPECIFIC COMMENTS

  1. Note references 1 & 7 appear to be identical (Borghi & Cimatti2010).

Reply: Thank you. We have made the correction.

  1. Please review the text on pp. 3-4 since it is confusing which novel task is intended to be first, second, third, fourth. For example, the fourth task is named immediately after the second task.

Reply: I have added a statement that the correct order of the tasks is shown in Appendix A, the scoring guide.

Round 2

Reviewer 1 Report

The authors made significant improvements and the paper is acceptable in its present form.